# Voxelmorph++

## Going beyond the cranial vault with keypoint supervision and multi-channel instance optimisation

Mattias P. Heinrich[1][0000−0002−7489−1972] and
Lasse Hansen[1][0000−0003−3963−7052]

Institute of Medical Informatics, University of Lübeck, Germany
{heinrich,hansen}@imi.uni-luebeck.de

**Abstract.** The majority of current research in deep learning based image registration addresses inter-patient brain registration with moderate deformation magnitudes. The recent Learn2Reg medical registration benchmark has demonstrated that single-scale U-Net architectures, such as VoxelMorph that directly employ a spatial transformer loss, often do not generalise well beyond the cranial vault and fall short of state-of-the-art performance for abdominal or intra-patient lung registration. Here, we propose two straightforward steps that greatly reduce this gap in accuracy. First, we employ keypoint self-supervision with a novel network head that predicts a discretised heatmap and robustly reduces large deformations for better robustness. Second, we replace multiple learned fine-tuning steps by a single instance optimisation with hand-crafted features and the Adam optimiser. Different to other related work, including FlowNet or PDD-Net, our approach does not require a fully discretised architecture with correlation layer. Our ablation study demonstrates the importance of keypoints in both self-supervised and unsupervised (using only a MIND metric) settings. On a multi-centric inspiration-exhale lung CT dataset, including very challenging COPD scans, our method outperforms VoxelMorph by improving nonlinear alignment by 77% compared to 19% - reaching target registration errors of 2 mm that outperform all but one learning methods published to date. Extending the method to semantic features sets new stat-of-the-art performance on inter-subject abdominal CT registration.

**Keywords:** registration · heatmaps · deep learning.

## 1 Introduction

Medical image registration aims at finding anatomical and semantic correspondences between multiple scans of the same patient (intra-subject) or across a population (inter-subject). The difficulty of this task with plentiful clinical applications lies in discriminating between changes in intensities due to image appearance changes (acquisition protocol, density difference, contrast, etc.) and nonlinear deformations. Advanced similarity metrics may help in finding a good

contrast-invariant description of local neighbourhoods, e.g. normalised gradient fields [5] or MIND [12]. Due to the ill-posedness of the problem some form of regularisation is often employed to resolve the disambiguity between several potential local minima in the cost function. Powerful optimisation frameworks that may comprise iterative gradient descent, discrete graphical models or both (see [24] for an overview) aim at solving for a global optimum that best aligns the overall scans (within the respective regions of interest). Many deep learning (DL) registration frameworks (e.g. DLIR [26] and VoxelMorph [1]) rely on a spatial transformer loss that may be susceptible to ambiguous optimisation landscapes - hence multiple resolution or scales levels need to be considered. The focus of this work is to reflect such local minima more robustly in the loss function of DL-registration. We propose to predict probabilistic displacement likelihoods as heatmaps, which can better capture multiple scales of deformation within a single feed-forward network.

**Related work:** Addressing large deformations with learning based registration is generally approached by either multi-scale, label-supervised networks [13, 20, 19] or by employing explicitly discretised displacements [7, 9]. Many variants of U-Net like architectures have been proposed that include among others, dual-stream [16], cascades [28] and embeddings [3]. Different to those works, we do neither explicitly model a discretised displacement space, multiple scales or warps, nor modify the straightforward feed-forward U-Net of DLIR or VoxelMorph.

Point-cloud registration is another research field of interest (FlowNet3d [18]) that has however so far been restricted to lung registration in the medical domain [6]. Stacked hourglass networks that predict discretised heatmaps of well-defined anatomical landmarks are commonly used in human pose estimation [22]. They are, however, restricted to datasets and registration applications where not only pairwise one-to-one correspondences can be obtained as training objective but generic landmarks have to be found across all potential subjects. Due to anatomical variations this restriction often prevents their use in medical registration. Combining an initial robust deformation prediction with instance optimisation or further learning fine-tuning steps (Learn-to-optimise, cf. [25]) has become a new trend in learning-based registration, e.g. ([23, 14]). This paradigm, which expects coarse but large displacements from a feed-forward prediction, can shift the focus away from sub-pixel accuracy and towards avoiding failure cases in the global transformation due to local minima. It is based on the observation that local iterative optimisers are very precise when correctly initialised and have become extremely fast due to GPU acceleration.

**Contributions:** We demonstrate that a single-scale U-Net without any bells and whistles in conjunction with a fast MIND-based instance optimisation can achieve or outperform state-of-the-art in large-deformation registration. This is achieved by focussing on coarse scale global transformation by introducing a novel heatmap prediction network head. In our first scenario we employ weak self-supervision, through automatic keypoint correspondences [10]. Here, the

heatmap enables a discretised integral regression of displacements to directly and explicitly match the keypoint supervision. Second, we incorporate the heatmap prediction into a non-local unsupervised metric loss. This enables a direct comparison within the same network architecture to the commonly used spatial transformer (warping) loss in unsupervised DL registration and highlights the importance of providing better guidance to avoid local minima. Our extensive ablation experiments with and without instance optimisation on a moderately large and challenging inspiration-exhale lung dataset demonstrate state-of-the-art performance.

Our code is publicly available at:
https://www.github.com/mattiaspaul/VoxelMorphPlusPlus

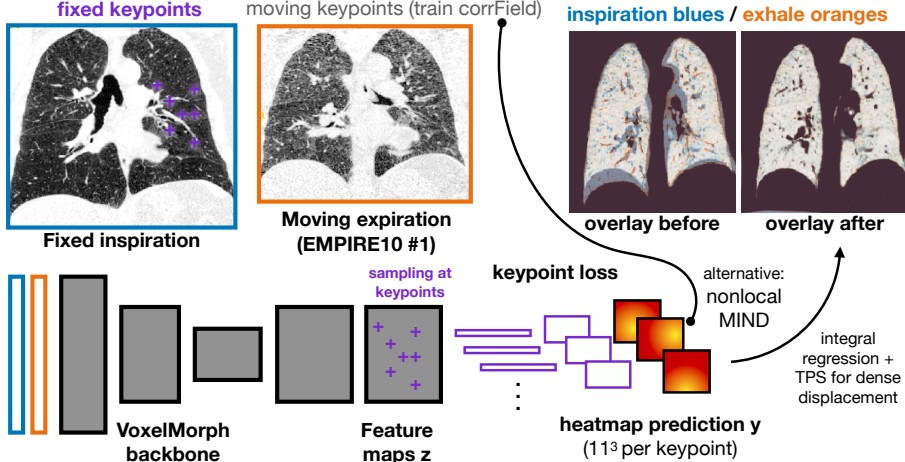

**Fig. 1.** Overview of method and qualitative result for held-out case #1 from [21]. The key new element of our approach is the heatmap prediction head that is appended to a standard VoxelMorph. It helps overcome local minima through a probabilistic loss with either automatic keypoint correspondences or non-locally weighted MIND features.

## 2    Method

Keypoint correspondences are an excellent starting point to explore the benefits of incorporating a heatmap prediction within DL-registration. Our method can be either trained when those automatically computed displacements at around $|K| \approx 2000$ locations per scan are available for training data, or we can use a non-local unsupervised metric loss (see details below). In both scenarios, we use Förstner keypoints in the fixed scan to focus on distinct locations within a region of interest. We will first describe the baseline single-scale U-Net backbone, followed by our novel heatmap prediction head and the non-local MIND loss (which is an extension of [11] to 3D).

**Baseline backbone:** Given two input CT scans, $F \rightarrow \mathbb{R}^3$ fixed image and $M \rightarrow \mathbb{R}^3$ moving image and a region of interest $\Omega \in \mathbb{R}^3$, we firstly define a feed-forward U-Net [4] $\Theta(F, M, \Omega, \theta) \rightarrow \mathbb{R}^C$ with trainable parameters $\theta$ that maps the concatenated input towards a shared $C$-dimensional feature representation $\mathbf{z}$ (we found C≈64 is expressive enough to represent displacements). This representation may have a lower spatial resolution than $F$ or $M$ and is the basis for predicting a (sparse) displacement field $\varphi$ that spatially aligns $F$ and $M$ within $\Omega$. $\Theta$ comprises in our implementation a total of eleven 3D convolution blocks, each consisting of a $3 \times 3 \times 3$ convolution, instance normalisation (IN), ReLU, a $1 \times 1 \times 1$ convolution, and another IN+ReLU. Akin to VoxelMorph, we use $2 \times 2 \times 2$ max-pooling after each of the four blocks in the encoder and nearest neighbour upsampling to restore the resolution in the decoder, but use a half-resolution output. The network has up to $C = 64$ hidden feature channels and 901'888 trainable parameters.

Due to the fact that this backbone already contains several convolution blocks on the final resolution at the end of the decoder, it is directly capable of predicting a continuous displacement field $\varphi$ by simply appending three more $1 \times 1 \times 1$ convolutions (and IN+ReLU) with a number of output channels equal to 3.

**Discretised heatmap prediction head:** The aim of the heatmap prediction head is to map a $C$-dimensional feature vector (interpreted as a $1 \times 1 \times 1$ spatial tensor with $|K|$ being the batch dimension) into a discretised displacement tensor $\mathbf{y} \in \mathcal{Q}$ with predefined size and spatial range $R$ (see Fig. 1). Here we chose $R = 0.3$, in a coordinate system that ranges from $-1$ to $+1$, which captures even large lung motion between respiratory states. We set $\mathcal{Q} \rightarrow \mathbb{R}^{11 \times 11 \times 11}$ to balance computational complexity and limit quantisation effects. This means we need to design another *nested* decoder that increases the spatial resolution from 1 to 11. Our heatmap network comprises a transpose 3D convolution with kernel size $7 \times 7 \times 7$, six further 3D convolution blocks (kernel size $7 \times 7 \times 7$ and IN+ReLU) once interleaved with a single trilinear upsampling to $11 \times 11 \times 11$. It has 462'417 trainable parameters and its number of output channels is equal to 1.

Next, we can define a probabilistic displacements tensor $\mathcal{P} \rightarrow \mathbb{Z}^6$ using a softmax operation along the combined displacement dimensions as:

$$\mathcal{P}(\mathbf{x}, \Delta\mathbf{x}) = \frac{\exp(y(\mathbf{x}, \Delta\mathbf{x}))}{\sum_{\Delta\mathbf{x}} \exp(y(\mathbf{x}, \Delta\mathbf{x}))}, \tag{1}$$

where $\mathbf{x} \rightarrow \mathbb{Z}^3$ are global spatial 3D coordinates and $\Delta\mathbf{x} \rightarrow \mathbb{Z}^3$ local 3D displacements. In order to define a continuous valued displacement field, we apply a weighted sum:

$$\varphi(\mathbf{x}) = \sum_{\Delta\mathbf{x}} \mathcal{P}(\mathbf{x}, \Delta\mathbf{x}) \cdot \mathcal{Q}(\Delta\mathbf{x}) \tag{2}$$

This output is used during training to compute a mean-squared error between predicted and pre-computed keypoint displacements. Since, the training correspondences are regularised using a graphical model, we require no further penalty.

**Non-local MIND loss:** To avoid the previously described pitfalls of directly employing a spatial transformer (warping) loss, we can better employ the probabilistic heatmap prediction and compute the discretely warped MIND vectors of the moving scan implicitly by a weighted average of the underlying features within pre-defined capture region (where $c$ describes one of the 12 MIND channels) as:

$$\mathrm{MIND}_{\mathrm{warped}}(c, \mathbf{x}) = \sum_{\Delta \mathbf{x}} \mathcal{P}(\mathbf{x}, \Delta \mathbf{x}) \cdot \mathrm{MIND}(c, \mathbf{x} + \Delta \mathbf{x}) \tag{3}$$

.

**Implementation details:** Note that the input to both the small regression network (baseline) and our proposed are feature vectors sampled at the keypoint locations, which already improves the baseline architecture slightly. We use trilinear interpolation in all cases where the input and output grids differ in size to obtain off-grid values. All predicted sparse displacements $\varphi$ are extrapolated to a dense field using thin-plate-splines with $\lambda = 0.1$ that yields $\varphi^*$.

For the baseline regression setup (VoxelMorph) we employ a common MIND warping loss and a diffusion regularisation penalty that is computed based on the Laplacian of a kNN-graph ($k = 7$) between the fixed keypoints. The weighting of the regularisation was empirically set to $\alpha = 0.25$. We found that using spatially aggregated CT and MIND tensors the former using average pooling with kernel size 2, the latter two of those pooling steps, leads to stabler training in particular for the regression baseline.

**Multi-channel instance optimisation:** We directly follow the implementation described in [23][1]. It is initialised with $\varphi^*$, runs for 50 iterations, employs a combined B-spline and diffusion regularisation coupled with a MIND metric loss and a grid spacing of 2. This step is extremely fast, but relies on a robust initialisation as we will demonstrate in our experiments. The method can also be employed when semantic features, e.g. segmentation predictions from an nnUNet [17], are available in the form of one-hot tensors.

## 3  Experiments and Results

We perform extensive experiments on inspiration-exhale registration of lung CT scans - arguably one of the most challenging tasks in particular for learning-based registration [14]. A dataset of 30 scan pairs with large respiratory differences is collected from EMPIRE10 (8 scan pairs #1, #7, #8, #14, #18, #20, #21 and #28) [21], Learn2Reg Task 2 [15] and DIR-Lab COPD [2] (10 pairs). The exhale and scans are resampled to $1.75 \times 1.25 \times 1.75$ mm and $1.75 \times 1.00 \times 1.25$ mm respectively to account for average overall volume scaling and a fixed region with dimensions $192 \times 192 \times 208$ voxels was cropped that centres the mass of automatic lung masks. Note that this pre-processing approximately halves the initial target registration error (TRE) of the COPD dataset. Lung masks are

---

[1] https://github.com/multimodallearning/convexAdam

also used to define a region-of-interest for the loss evaluation and to mask input features for the instance optimisation. We split the data into five folds for cross-validation that reflect the multi-centric data origin (i.e. approx. two scans per centre are held out for validation each).

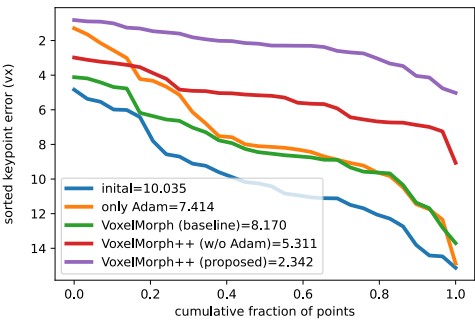

**Fig. 2.** Cumulative keypoint error of proposed model compared to a VoxelMorph baseline and using only Adam instance optimisation with MIND.

| | UNet | Heatmap | Keypoints | Error w/o Adam | Error w/ Adam |
|---|---|---|---|---|---|
| Initial / Adam | | | | 10.04 vx | 7.41 vx |
| VoxelMorph | ✓ | | | **8.17 vx** | 4.80 vx |
| VM + Heatmap | ✓ | ✓ | | 6.49 vx | 3.18 vx |
| VM + Keypoints | ✓ | | ✓ | 6.30 vx | 2.79 vx |
| **VoxelMorph++** | ✓ | ✓ | ✓ | 5.31 vx | **2.34 vx** |

**Table 1.** Results of ablation study on lung CT: VoxelMorph++ improves error reduction of nonlinear alignment from 18% to 77%.

**Keypoint Self-supervision:** To create correspondence as self-supervision for our proposed VoxelMorph++ method, we employ the **corrField**[10][2], which is designed for lung registration and based on a discretised displacement search and a Markov random field optimisation with multiple task specific improvements. It runs within a minute per scan pair and creates $\approx 2000 = |K|$ highly accurate ($\approx 1.68$ mm) correspondences at Förstner keypoints.

As additional experiment we extend our method to the inter-subject alignment of 30 abdominal CTs [27] that was also part of the Learn2Reg 2020 challenge (Task 3) and provides 13 difficult anatomical organ labels for training and validation.

**Ablation study:** We consider a five-fold cross-validation for all ablation experiments with an initial error of 10.04 vx (after translation and scaling) across 30 scan pairs computed based on keypoint correspondences. Employing only the Adam instance optimisation with MIND features results in an error of 8.17 vx, with default settings of grid spacing = 2 voxels, 50 iterations and $\lambda_{Adam} = 0.65$. Note that a dense displacement is estimated with a parametric B-spline model. We start from the slightly improved VoxelMorph **baseline** with MIND loss, diffusion regularisation and increased number of convolution operations described above. This yields a keypoint error of **8.17 vx** that represents an error reduction of 19% and can be further improved to 4.80 vx when adding instance optimisation. A weighting parameter $\lambda = 0.75$ for diffusion regularisation was empirically found with $k = 7$ for the sparse neighbourhood graph of keypoints. Replacing

---

[2] http://www.mpheinrich.de/code/corrFieldWeb.zip

the traditional spatial transformer loss with our proposed heatmap prediction head that uses the nonlocal MIND loss much improves the performance to 6.49 vx and 3.18 vx (with and without Adam respectively). But the key improvement can be gained when including the self-supervised keypoint loss. Using our baseline VoxelMorph architecture that regresses continuous 3D vectors, we reach 6.30 and **2.79 vx**. See Table 1 and Fig. 2 for numerical and cumulative errors. Our heatmap-based network and the instance optimisation require around 0.43 and 0.41 seconds inference time, respectively. The complexity of transformations measured as standard deviation of log-Jacobian determinants is on average 0.0554. The number of negative values is zero (no folding) in 8 out of 10 COPD cases and negligible ($< 10^{-4}$) in the others.

**Comparison to state-of-the-art:** When evaluation the target registration error (TRE) in mm for the 10 pairs of DIR-Lab COPD [2] one of the most challenging benchmarks in medical registration with an initial misalignment of 23.36 mm (and 12.0 mm after pre-alignment), we reach 2.16 mm. This compares very favourable to VoxelMorph+ with 7.98mm and LapIRN with 4.76mm. Of all published DL-methods only GraphRegNet [7] is superior with 1.34 mm. The high visual quality of our registration is shown in Fig. 3.

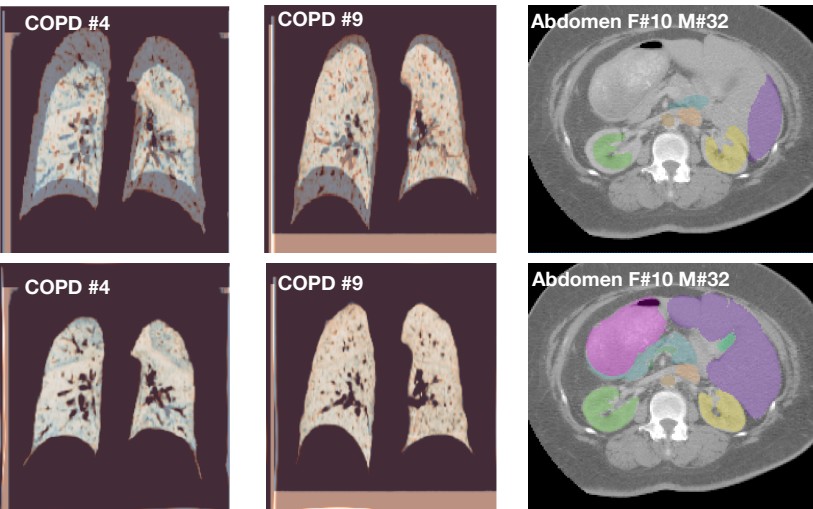

**Fig. 3.** Exemplary results of our proposed method before (top row) and after (bottom row) registration. Fixed exhale scans are shown in blue and inspiration in orange shades respectively (adding up to grayscale when aligned). For abdominal alignment transformed segmentation labels are shown, here: right kidney ■, left kidney ■, gallbladder ■, liver ■, stomach ■, aorta ■ and pancreas ■ are visible.

**Limitations and further potential:** We have not yet considered more advanced network architectures as backbone, e.g. two-stream or multi-level, which

are likely to yield further improvements. However, based on our experiments we expect that it could merely reduce the reliance on instance optimisation.

**Inter-subject abdominal CT registration:** We apply our proposed VoxelMorph++ model with nonlocal loss and no architectural modifications to another challenging task of inter-subject abdominal CT registration with initial Dice overlap for 13 organs of only 25.9% and weakly supervised learning (45 registration pairs). Following [8], we decouple the semantic feature extraction and directly train an nnUNet model [17]. The best published VoxelMorph model that was trained with label-supervision and extended to a two-stream architecture reached 43.9% [23], the two top-ranked methods of the Learn2Reg challenge yield 65.7% (ConvexAdam [23]) and 67% (LapIRN [20]) respectively. Directly employing instance optimisation with 25 iterations on the nnUNet features achieves 62.9%. We use 2048 keypoints that are sampled inversely proportional to the predicted label maps and employ two warps (and inverse consistency for the first of them). Our model substantially outperforms VoxelMorph with 52.3% and sets a new state-of-the-art performance after instance optimisation reaching 69.6% with a total run time of less than a second.

## 4    Discussion and conclusions

: Our results demonstrate that contrary to previous belief, a simple single-scale U-Net architecture can provide large deformation estimation that is robust enough to reach high accuracy with a subsequent instance optimisation. The key insight of our work is the importance to predict a discretised heatmap to alleviate the problematic direct regression and use strong self-supervision either using automatic keypoint correspondences or a nonlocal multichannel loss together with a straightforward instance optimisation. Our work is related to mlVIRN [13], which also incorporates a keypoint loss for lung registration in addition to lobe segmentations, but has to be trained with several hundreds of paired CTs and did not report TRE values for DIRlab-COPD. Our network can be trained within 17 minutes on a single RTX A4000 requiring less than 2 GByte of VRAM, indicating the improved training efficiency with fewer scans when using heatmaps. GraphRegNet [7] is similar in that it also employs heatmaps (integral regression) but more explicitly by defining the exact same discretised displacement grid beforehand and computing an SSD cost tensor based on hand-crafted features as input. While it outperforms our method with a TRE of 1.34mm it appears to be more tailored towards the specific task and might not be easily extendable to end-to-end feature learning or abdominal registration.

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
