# OpenReview forum: "Voxelmorph++ Going beyond the cranial vault with keypoint supervision and multi-channel instance optimisation"
_WBIR.info/2022/Workshop/Biomedical_Imaging_Registration — WBIR 2022_

### Official Review · Reviewer_GuF6 · 2022-02-11

**Rating:** 5
**Confidence:** 4
**Recommendation:** Long Oral

**Deanonymize Review:**

no

**Detailed Comments:**

-	p 5 The exhale and scans: word missing

**Paper Type:**

methodological development

**Strengths Weaknesses:**

The paper addresses the problem of many deep learning methods to image registration that large deformations cannot be recovered well. Although several solutions have been proposed, this is still a timely and important topic. The method presented is based on a very different approach to what has been suggested up till now, using key points from features. The method is demonstrated on inspiration-expiration lung CT and on abdominal CT (interpatient), with good results.

Strengths:
-	timely and interesting topic
-	approach is different from published ones and could have additional advantages
-	good results

Weaknesses:
-	comparison to other methods excludes some essential ones
-	comparison hampered by different training setup

The technique presented is interesting because of its different perspective on the problem, using a single scale. The results are good, although outperformed by some approaches. However, I expect this method based on key points to possess advantageous properties over other methods. This could well turn out to be a preferred technique for specific applications.
The use of the feature detector does make this method somewhat slower than various other networks.
The comparison with other methods is very useful. However, methods have been left out, methods that are more relevant than those included. See the references below for two examples. Both these methods are evaluated on the same data and yield results closer to those of the proposed method than the methods that are included. More importantly, both these methods were trained on other data, whereas in this paper a cross-validation is used. It is likely the methods achieve similar results when applied in a similar setting.

Sang and Ruan, Scale-adaptive deep network for deformable image registration, Med. Phys. 48(7), 2021
Eppenhof et al., Progressively trained convolutional neural networks for deformable image registration, Med. Image Anal. 39(5) 2020

---

### Official Review · Reviewer_3cPL · 2022-02-20

**Rating:** 2
**Confidence:** 5

**Deanonymize Review:**

no

**Detailed Comments:**

The title of the paper is misleading should be changed. The paper talks about lung registration which is not mentioned in the title and not cranial vault registration.

**Paper Type:**

methodological development

**Strengths Weaknesses:**

Strengths:

This work proposes to use deep learning to build a landmark/feature-based image registration algorithm for the purpose of registering lung image volumes.

Weaknesses:

There is little novelty in this paper.

There literature review does not mention much of the prior point-based image registration work. There have been many medical imaging point-cloud registration methods developed over the last three decades. The authors should cite the seminal iterative closest point (ICP) algorithm by Besl and McKay 1992 and then modify the following statement "Point-cloud registration is another research field of interest (FlowNet3d [18]) that has however so far been restricted to lung registration in the medical domain [6]." Point cloud image registration has been a part of the Analyze software (Mayo Clinic) since the 1990s. Anand Rangarajan is one of the pioneers of point-based nonrigid medical image registration. Here are a couple of citations to his work. A new point matching algorithm for non-rigid registration. H Chui, A Rangarajan - Computer Vision and Image Understanding, 2003. The softassign Procrustes matching algorithm. A Rangarajan, H Chui, FL Bookstein -IPMI 1997. Here is a web page that lists a lot of landmark-based medical image registration papers https://www.science.gov/topicpages/l/landmark-based+image+registration.html

This algorithmic approach in this paper seems similar in spirit to the following paper. DRAMMS: Deformable registration via attribute matching and mutual-saliency weighting.Yangming Ou, Aristeidis Sotiras, Nikos Paragios, Christos Davatzikos. Medical Image Analysis, 15(4): 622-639, 2011. The authors should compare their method to the DRRAMS algorithm https://www.med.upenn.edu/sbia/dramms.html

---

### Official Review · Reviewer_KUrr · 2022-02-21

**Rating:** 4
**Confidence:** 5
**Recommendation:** Short Oral

**Deanonymize Review:**

no

**Detailed Comments:**

    • Overall, I find the presentation of the experimental results section a bit confusing. I think the authors should reorganize their results, summarizing them in tables. And clearly mention their evaluation protocol for all the methods.
    • Does the number of keypoint need to be fixed during training and testing? If yes, how does the user decide about it?
    • I think the authors should include some more description and discussion for Fig.2.


**Paper Type:**

both

**Strengths Weaknesses:**

Strengths

    • The method is interesting and the keypoint correspondences seem to boost the performance of the registration framework.
    • The methodology sounds and is easy to follow.
    • The paper includes discussion and comparison with other state-of-the-art methods.
    • The authors include their implementation and code in the submission.

Weaknesses

    • The authors claim that their design is suitable for large deformations on which the spatial transformer loss may give suboptimal solutions. I think that this claim is not properly supported by the evaluation and the quantitative/ qualitative results. Some more discussion and comparison is needed to prove the good performance on large deformations with respect to learning based methods that use the spatial transformer.
    • The authors present an ablation study for the different components of their method mainly in terms of keypoint errors (Table 1). Even if the authors discuss the performance of their method with respect to standard deviation log-Jacobian in the manuscript and dice (for the abdominal CT dataset), I think they should perform an extended evaluation and comparison of their method to prove the superiority of their methodological components. In particular, the dice metric and the standard deviation log-Jacobian should be included in the ablation that is summarised in Table1 and all metrics should be included on the comparison to state-of-the-art.

---

### Decision · Program_Chairs · 2022-02-22

Accept